# The Association between Alcohol Drinking Patterns and Health-Related Quality of Life in the Korean Adult Population: Effects of Misclassification Error on Estimation of Association

**DOI:** 10.3390/ijerph17217758

**Published:** 2020-10-23

**Authors:** Eun Sook Lee, Boyoung Kim

**Affiliations:** 1Department of Nursing, Gyeongnam National University of Science and Technology, Jinju, Gyeongsangnamdo 52725, Korea; eslee5335@gntech.ac.kr; 2College of Nursing, Institute of Health Science, Gyeongsang National University, Jinju, Gyeongsangnam-do 52727, Korea

**Keywords:** alcohol drinking, quality of life, AUDIT, gender

## Abstract

The purpose of this study was to investigate the association between drinking patterns and health-related quality of life (HRQoL) in the Korean general population and to validate the estimation of the association of alcohol use on HRQoL when former drinkers are separated from never drinkers and low-risk drinkers depending on gender. Data were collected from 23,055 adults (over 19 years old) who completed the Korean National Health and Nutritional Examination Survey (2010–2013). Multivariate logistic and linear regression analyses were performed to investigate the association between drinking patterns and HRQoL. When former drinkers were separated from never drinkers and low-risk drinkers to control for misclassification bias, there were gender differences in the associations between alcohol use and HRQoL. Although the estimation of the association of alcohol use was not valid in men, the estimation of association was valid in women, as low-risk women drinkers showed better HRQoL than nondrinkers. Therefore, when conducting research related to alcohol and health, analyses should correct for the various confounding variables and minimize the misclassification bias of drinking patterns. It is also necessary to consider gender characteristics when intervening to improve HRQoL related to drinking.

## 1. Introduction

Harmful alcohol use is ranked as the 7th leading risk factor for premature death and disability worldwide [1] and has been reported to cause significant health deterioration as a major risk factor for about 200 acute and chronic diseases [2,3]. However, low-to-moderate alcohol use has been suggested to have beneficial effects on specific health conditions, such as ischemic heart disease, ischemic stroke, diabetes mellitus, and total mortality [3,4,5,6]. As such, the association between alcohol use and health seems complicated.

This is similar to the association between alcohol and health-related quality of life (HRQoL). HRQoL refers to the areas of life directly affected by changes in health and is a multi-dimensional concept that covers significant domains of daily functioning and subjective experiences, such as physical functioning, social role functioning, somatic sensation, and subjective well-being [7]. Measuring HRQoL can help determine the burden of preventable diseases, injuries, and disabilities and enables the examination of the relationship between HRQoL and risk factors [8].

For these reasons, general population-based studies on alcohol consumption and HRQoL have previously been conducted to evaluate alcohol-related health, but the results have been inconsistent. Many studies have reported an inverted U- or J-shaped relationship between alcohol consumption and HRQoL [9,10,11,12], showing that the HRQoL of light and moderate drinkers is better than that of nondrinkers and heavy drinkers, while other studies have shown that moderate and heavy drinkers had better HRQoL than nondrinkers [13,14,15], and still others found no difference in HRQoL between nondrinkers, moderate drinkers, and heavy drinkers [16,17]. This inconsistency may be due to differences in the subject populations, classification of drinking patterns, cultural perceptions of drinking, and corrected confounding variables [9,13,14,16].

Meanwhile, criticisms have been constantly raised on the methodology of epidemiological studies that found protective effects of alcohol consumption. Various selection biases may have led to a systematic overestimation of the protective effects of “moderate” alcohol consumption on mortality [18]. Fillmore et al. [19] reported that a systemic misclassification error may be present in most prospective studies on alcohol use and mortality risk, which is caused by including former or occasional drinkers in the abstainer category, and pointed out that abstainers and light or moderate drinkers are at equal risk of coronary heart disease. According to the “sick quitter hypothesis”, individuals are likely to quit or reduce drinking because of old age or illness [20]. Therefore, including former or occasional drinkers in the abstainer category can disrupt the protective effect of light or moderate drinking, as the health profile of the abstainer category is poor [19,21]. A meta-analysis performed by excluding former and occasional drinkers showed a significantly weaker protective effect of moderate alcohol consumption on total mortality [5]. In addition, meta-analyses [19,22] that explored the effects of abstainer reference group bias and various confounding variables by controlling for former drinker bias and other study characteristics found no evidence of the reduced risk of all-cause mortality in low-volume drinkers and, also, observed no J-shaped curves. This is similar to the relationship between alcohol consumption and HRQoL. In a study that separated former drinkers and other abstainers [17], former drinkers had poorer quality of life than other abstainers, and moderate alcohol use had no misclassification error on the quality of life. Meanwhile, Liang et al. [23] argued that excluding former drinkers from the drinker group exaggerates the differences in health status between abstainers and current drinkers and that former drinkers should be assigned to the drinker category based on their previous alcohol consumption patterns. Therefore, it is necessary to control for former drinkers and classify drinking patterns accordingly to minimize potential bias when studying the association between drinking and health outcomes.

Previous studies that examined the relationship between drinking and HRQoL using representative population-based data in Korea did not control for former drinkers and reported that nondrinkers or nonproblematic drinkers, including former drinkers, had lower HRQoL levels than problematic drinkers [9,24,25,26]. This hypothesis that moderate levels of alcohol consumption provide a benefit to HRQoL may influence the development of public health policies and national alcohol use guidelines to reduce the harm caused by alcohol use. In a recent global burden of disease (GBD) report on alcohol use [1], it was found that, when the confounding factors of different reference categories were controlled for in a meta-analysis to evaluate the risk of alcohol use, the level of alcohol consumption that minimized health loss was zero. Therefore, this report suggested that alcohol control policies may need to be revised worldwide to focus the efforts on lowering the overall population-level consumption. Although Korea has implemented health policies to prevent alcohol abuse, the alcohol consumption per capita per year in adults over 15 years old has increased from 9.9 L in 2010 to 10.2 L in 2016 (men: 16.7 L and women: 3.9 L), which is higher than the global average (6.4 L) [2]. In particular, the monthly heavy drinking rate and high-risk alcohol consumption rate of Korean adult men were 52.7% and 21.0%, respectively, which were significantly higher than those of women (25.0% and 7.2%) [27]. Although men experience more deaths and disabilities caused by high alcohol consumption as compared to women [2], men have a higher level of HRQoL than women [9,10,14,24], implying a possible gender difference in the association between drinking and HRQoL.

The purpose of this study was to examine the association between drinking patterns and HRQoL in adults (over 19 years old) and investigate whether there are gender differences in this association by using representative population-based data from Korea. In particular, this study aimed to consider the effects of misclassification error in the relationship between drinking patterns and HRQoL when former drinkers are separated from never drinkers and low-risk drinkers. Based on these results, the study provides a basis for public health guidance and policy development regarding drinking.

## 2. Materials and Methods

### 2.1. Study Design

This was a cross-sectional study to investigate the association between drinking patterns and HRQoL in Korean adults.

### 2.2. Data Setting and Study Sample

This study conducted a secondary analysis using raw data from the Korean National Health and Nutritional Examination Survey (KNHANES) from 2010 to 2013. KNHANES used a two-stage stratified cluster sampling method with households and housing units (from the Population and Housing Census District) as the primary and secondary sampling units to select the survey targets representing Korea. In the 5th survey, 25,533 individuals (80.8%) out of 31,596 participated (2010–2012) [28] and 8018 (79.3%) out of 10,113 participated in the 1st year of the 6th survey [29]. This study selected 23,055 adults over the age of 19 (*n* = 25,422) who completed the Alcohol Use Disorder Identification Test (AUDIT) and EuroQol 5-Dimensions (EQ-5D) questionnaire as the sample for the final analysis, from a total of 33,113 participants in the 2010–2013 health surveys.

### 2.3. Measurements

#### 2.3.1. Alcohol Drinking Patterns

Drinking patterns were identified using a self-report questionnaire related to alcohol consumption and the AUDIT scores. AUDIT is a tool developed by the WHO to screen for dangerous and harmful patterns of alcohol consumption and consists of 10 questions in three domains (hazardous alcohol use, dependence symptoms, and harmful alcohol use) [30]. The score ranges from 0 to 40, with respondents answering questions on a 4-point Likert scale. Higher scores indicate higher levels of problematic drinking. AUDIT scores of 8 or above are classified as hazardous drinking (or risky drinking), 16 or above as harmful drinking, and 20 or above as alcohol dependence [30].

In this study, those who answered “no” to the first question related to drinking, “Have you ever had more than one drink of alcohol in your life?” were classified as never drinkers. The participants who answered “yes” to this question and reported not consuming any alcohol within the last year (AUDIT score: 0) were classified as former drinkers. Participants with AUDIT scores of 1–7 were classified as low-risk drinkers, 8–15 as hazardous drinkers, 16–19 as harmful drinkers, and 20 or above as having alcohol dependence.

#### 2.3.2. Health-Related Quality of Life

This study used EQ-5D to measure the HRQoL. The EQ-5D is a standardized tool for measuring self-reported general health status, developed by the EuroQol Group [31]. The tool consists of five dimensions: mobility, self-care, usual activities, pain/discomfort, and anxiety/depression, and each dimension has three levels: level 1 (no problems), level 2 (some problems), and level 3 (extreme problems). There is no official cutoff value in the EQ-5D index score that indicates a reduced or impaired health-related quality of life. The EQ-5D index was calculated by applying weights to the measurements of each dimension of the EQ-5D, and this study measured the scores calculated by applying the weights derived from a Korean validation study [28,32]. Thus, “1” means completely healthy, “0” refers to death, and a score below 0 indicates a health status worse than death.

#### 2.3.3. Sociodemographic and Health-Related Characteristics

The sociodemographic variables were gender (male and female); age (years); personal income level (low, lower-middle, upper-middle, and high); education level (elementary school or below, middle school, high school, and college graduate or above); marital status (married/cohabiting, single, separated, widowed, and divorced); and employment status (employed or unemployed). Health-related variables included self-reported smoking, physical activity, obesity, and chronic disease morbidity. Respondents were classified regarding smoking status as ”nonsmokers”, “former smokers”, and “current smokers”, depending on whether they were lifelong smokers or current smokers. Physical activity was divided into intensive, moderate, and low in terms of practicing 1 or more walking hours per week. Respondents with BMI ≥ 25 kg/m^2^ were classified as obese. Patients who were diagnosed with one or more of the following diseases were classified as being in the chronic disease morbidity group: circulatory disease (hypertension, dyslipidemia, or stroke); musculoskeletal disease (osteoarthritis or rheumatoid arthritis); respiratory disease (pulmonary tuberculosis or asthma); liver disease (hepatitis B or C or hepatocirrhosis); cancer (stomach, liver, colon, breast, cervix, lung, or thyroid cancer); and diabetes mellitus.

### 2.4. Data Analysis

PASW Statistics 20 (SPSS Inc., Chicago, IL, USA) was used to analyze the data. A complex-sample analysis was performed based on an analysis plan file in which weights, stratification variables, and primary sampling units were designed. The analysis variables included all missing data to exclude the possibility of bias in the variance estimator. In terms of participant characteristics, a frequency analysis was used to calculate the frequency, percentage, and standard error. The differences in participant characteristics and EQ-5D scores depending on the drinking patterns were calculated using the χ2 test and linear regression. A multivariate logistic regression was used to model the relationship between dichotomous reports of problems for each dimension of the EQ-5D and alcohol drinking patterns after controlling for other covariates. This study evaluated the EQ-5D index scores, HRQoL proportions, and adjusted odds ratios (ORs) and assessed each dimension of the EQ-5D and alcohol drinking patterns. This method may be inadequate for directly comparing the overall results consistently; thus, the evaluation was performed using the dimension of each EQ-5D item: level 1 (no problems), level 2 (some problems), and level 3 (extreme problems). In addition, multivariate linear regression was used to identify the relationship between alcohol drinking patterns and EQ-5D scores by gender after controlling for other covariates. Model 1 was developed by adjusting for age, gender, economic status, education level, marital status, and employment status. Model 2 was developed by adjusting for the factors listed in Model 1 in addition to smoking status, physical activity, obesity, and chronic disease. The α value was adjusted using the Bonferroni correction to compensate for multiple comparisons regarding the primary outcome measures. *p*-values were compared with this adjusted value to interpret and analyze the associations of the primary outcome measures.

## 3. Results

### 3.1. Characteristics of Study Participants by Alcohol Drinking Patterns

A total of 10.6% of the participants were never drinkers, 11.7% were former drinkers, 44.8% were low-risk drinkers, 21.0% were hazardous drinkers, 5.9% were harmful drinkers, and 6.0% were alcohol-dependent. The differences in age, gender, income level, education level, marital status, employment status, smoking history, physical activity, obesity, and chronic disease morbidity were significant depending on the drinking patterns (Table 1). The mean age of the whole sample was 45.4 years. The mean age of the never drinkers (58.2 years) was the highest and that of the hazardous drinkers (40.3 years) was the lowest (*p* < 0.001). Amongst the men, there was a higher proportion of hazardous and harmful drinkers and individuals with alcohol dependence, and amongst women, there was a higher proportion of never, former, and risky drinkers (*p* < 0.001). Compared to other drinking patterns, the ratio of low-income respondents was significantly higher in the never drinkers, former drinkers, and alcohol-dependent groups (*p* < 0.001), and the education levels were significantly lower in the same groups (*p* < 0.001). In terms of marital status, never drinkers and former drinkers showed higher ratios of widowed, divorced, and separated status, and most of the current drinkers were single (*p* < 0.001). The employment rate was higher in harmful drinkers and the alcohol-dependent group than in never drinkers and former drinkers (*p* < 0.001). In terms of health-related characteristics, the ratio of current smokers was higher in hazardous drinkers, harmful drinkers, and alcohol-dependent individuals than in never and former drinkers (*p* < 0.001), and the ratio of physically active respondents was also higher in hazardous and harmful drinkers (*p* < 0.001). The prevalence of obesity was the lowest in low-risk drinkers and the highest in harmful drinkers and individuals with alcohol dependence, showing a significant difference (*p* < 0.001). Chronic disease morbidity was significantly higher in never drinkers and former drinkers than in groups with other drinking patterns (*p* < 0.001) (Table 1).

### 3.2. Association of Alcohol Drinking Patterns with Compromised HRQoL

There was a significant difference in the ratio of “some and extreme problems” (L2 and L3) in the EQ-5D subdomains, depending on the drinking patterns (*p* < 0.001). The HRQoL impairment rate was the highest in never drinkers, followed by former drinkers, while hazardous and harmful drinkers showed the lowest impairment rates. Never drinkers also showed the lowest HRQoL, indicated by the EQ-5D index, followed by former drinkers, and the hazardous drinkers and harmful drinkers showed the highest HRQoL, with a significant difference (*p* < 0.001) (Appendix A).

Analysis of the association between impaired HRQoL and drinking patterns by dividing the EQ-5D subdomains into “no problems” (L1) and “problems” (L2 and L3) revealed that all of the drinking patterns had a significant association with impaired HRQoL in both model 1 (controlling for sociodemographic variables) and model 2 (with additional corrections for health-related variables). Compared to the never drinkers in model 2, former drinkers had significantly higher ORs for usual activities impairment (OR = 1.23; 95% confidence interval (CI) = 1.02 to 1.49) and pain/discomfort (OR = 1.24; 95% CI = 1.07 to 1.44), and the alcohol-dependent group had a significantly higher OR for pain/discomfort (OR = 1.31; 95% CI = 1.05 to 1.65). Meanwhile, low-risk and hazardous drinkers had significantly lower ORs (0.79 and 0.67, respectively) in self-care impairment compared to never drinkers (95% CI = 0.64 to 0.98 and 95% CI = 0.47 to 0.96, respectively). The former drinkers (OR = 1.30; 95% CI = 1.09 to 1.56), hazardous drinkers (OR = 1.29; 95% CI = 1.04 to 1.61), harmful drinkers (OR = 1.42; 95% CI = 1.03 to 1.94), and alcohol-dependent group (OR = 2.42; 95% CI = 1.86. 3.14) showed significantly higher ORs for anxiety/depression compared to never drinkers (Table 2).

### 3.3. Association of Alcohol Drinking Patterns with HRQoL by Gender

In terms of the HRQoL, according to drinking patterns by gender, the unadjusted means of the EQ-5D index of male former drinkers decreased dramatically compared to male never drinkers, increased gradually according to the level of drinking, and then tended to decrease again in alcohol dependence. For females, it was lowest in never drinkers, increased gradually according to the level of drinking, and then decreased again in alcohol dependence. The fully adjusted means of the EQ-5D index of males was highest in never drinkers and decreased in former drinkers and alcohol dependence. In the case of females, it decreased slightly in former drinkers compared to never drinkers, increased according to the level of drinking, and then decreased again in alcohol dependence (Table 3).

As for the association between alcohol drinking patterns and HRQoL by gender after correcting for sociodemographic and health-related variables, the EQ-5D index scores of male former drinkers were significantly lower (β = −0.018, 95% CI = −0.033 to −0.004) than those of male never drinkers, while the EQ-5D index scores of groups with other drinking patterns were lower but not to a statistically significant degree. In the case of women, the EQ-5D index scores of low-risk drinkers were significantly higher (β = 0.008, 95% CI = 0.001 to 0.015) than those of never drinkers, while the scores of former drinkers and the alcohol-dependent group were lower but not to a statistically significant degree (β = −0.003, 95% CI = −0.012 to 0.005 and β = −0.011, 95% CI = −0.039 to 0.017, respectively) (Table 4) (Figure 1).

## 4. Discussion

This was a population-based cross-sectional study to validate the estimation of the association of alcohol use with HRQoL based on the association analysis of drinking patterns and HRQoL when the misclassification error of former drinkers was controlled for.

First, the focus of this study was to validate the estimation of the association of alcohol use with HRQoL when former drinkers were separated from never drinkers and low-risk drinkers depending on gender. Population-based cross-sectional studies that did not control for former drinkers reported an inverted U- or J-shaped relationship between alcohol consumption and HRQoL [9,10,11,12] and suggested that light-to-moderate drinking has a misclassification error in HRQoL. However, in this study, there was no significant difference in EQ-5D index scores between male never drinkers and male low-risk and hazardous drinkers; thus, the misclassification error of the estimation of the association was not valid, and the overall shape of the curve showed a contrast between never drinkers and former drinkers. This is consistent with studies that found that separating former drinkers reduced the estimated benefits of light or moderate alcohol use against the mortality risk, resulting in no significant difference between long-term abstainers and drinkers [5,19,33]. However, in women, low-risk drinkers showed significantly higher EQ-5D index scores than never drinkers after correcting for covariates, which was different from men. Similar to this study, a Finnish population-based study [17] that separated former drinkers and examined gender differences also showed that all statistically significant associations between moderate alcohol use and quality of life disappeared in men when former drinkers were excluded and sociodemographic factors were controlled for. In addition, in the case of women, moderate alcohol use was associated with better self-rated health and higher EQ-5D scores compared to abstainers. In this study, 53.5% of women were low-risk drinkers (36.2% in men), and low-risk drinkers had a significantly lower mean age than never drinkers. Since a younger age is reported to be associated with a better HRQoL [24,34,35], for women in this study with lower drinking problems than men, the HRQoL of low-risk drinkers, who accounted for more than half of the women, was expected to be significantly higher. According to a cohort study that analyzed the association between alcohol consumption and mortality by stratifying gender and age groups (50–64 years and ≥65 years), the misclassification error of the estimation of the association of alcohol consumption was only seen in women drinkers aged 65 years or more when compared with never drinkers, while no misclassification error was observed in the other age groups [33]. This estimation of association may, however, be explained by the effect of selection biases across the age-gender strata; therefore, further analyses should be performed on the association between drinking patterns and HRQoL by stratifying by not only gender but, also, age groups.

Second, the former drinkers in this study showed a significant difference in HRQoL compared with never drinkers and low-risk drinkers. After correcting for the covariates in both genders, the former drinkers showed worse HRQoL than never drinkers, which is consistent with previous studies [17,35,36]. This supports the previous studies that brought up the “former drinker misclassification error” issue, showing that including former drinkers in the abstainer category can lead to an inflation of the estimate of abstaining effects on health outcomes. Previous meta-analyses on alcohol consumption and all-cause mortality [19,22,33] also showed that the mortality risk of former drinkers was higher than that of lifetime abstainers, and a study investigating the association between alcohol consumption and self-rated health [37] showed that former drinkers (≥one year of abstinence) had worse self-perceived health than never drinkers, as they reduced their alcohol consumption because of poor health. In this study, HRQoL differences between former, never, and low-risk drinkers were particularly significant in men compared to women. Men have higher levels of harmful alcohol use than women and more negative health outcomes related to drinking [9,10,38]. Therefore, men will likely have a higher incidence of sick quitters related to alcohol than women, which may result in worse HRQoL in former drinkers. Two studies conducted in Spain on populations of adults aged over 60 years [13] and adults aged over 18 years [37] reported no differences in physical and mental HRQoL between nondrinkers (including occasional drinkers) and ex-drinkers through a full-adjusted analysis. These different results can be attributed to the fact that the above studies did not distinguish based on gender and used different tools to classify drinking patterns and measure the HRQoL.

Third, 10.6% and 11.7% of Korean adults (over 19 years old) in this study were never drinkers and former drinkers, respectively. This was slightly different from a previous study on global alcohol use, where the proportions of Korean lifetime abstainers and former drinkers were 7.1% and 29%, respectively [2]. However, the previous study was based on a population over 15 years of age, and the two studies should thus be compared considering the age difference. The prevalence of alcohol use disorders (harmful drinking and alcohol dependence) among the participants in this study was 11.9%, which is similar to the previously reported 13.9% prevalence of alcohol use disorders in Koreans over 15 years of age. Meanwhile, the prevalence of alcohol use disorders in Korea was three times higher than the 4.7% prevalence reported in the WHO Western Pacific region, which shows the need to revise alcohol control policies to lower the total population-level consumption.

Fourth, the sociodemographic results showed that the heavy drinking group had higher ratios of lower age, male gender, higher education, higher income, single status, and being employed than never drinkers or former drinkers, which is consistent with previous research showing that heavy or high-risk drinking is associated with the male gender, younger age, higher education, higher income, single status, and being employed [38,39,40,41,42]. In addition, the prevalence of smoking and obesity was the highest in harmful drinkers and those with alcohol dependence, which was similar to previous studies investigating the association between negative health behaviors and heavy or high-risk drinking [9,38,39,41,42]. Although the causal relations between these health behaviors cannot be proven, they are all highly interrelated; thus, the focus should be on using health campaigns and education to address these issues in a more comprehensive manner [38,39]. Meanwhile, hazardous and harmful drinkers practiced more positive health behaviors, such as physical activity, and showed a lower prevalence of chronic diseases, which was unexpected. These results are similar to studies reporting higher levels of physical activity in heavy or high-risk drinkers [38,39] and the highest levels of participation in regular exercise in male alcohol use disorder groups and female hazardous drinking groups [9]. In general, the rate of physical activity decreases and the prevalence of chronic diseases increases with age [27]. In this study, the average ages of never drinkers (58.2 years) and former drinkers (50.5 years) were significantly higher than those of hazardous drinkers (40.3 years) and harmful drinkers (42.8 years); therefore, it is necessary to analyze the relationship between drinking patterns and health-related behaviors by dividing age groups in the future.

Finally, pain/discomfort was the most commonly reported impaired HRQoL dimension in all drinking patterns, followed by mobility impairment in never drinkers, former drinkers, and low-risk drinkers and anxiety/depression in hazardous drinkers, harmful drinkers, and alcohol-dependent individuals. The dimension with the lowest rate of impairment was self-care. This is consistent with a previous study on the HRQoL of the general population in Korea [34], where the EQ-5D dimension with the most problems reported was pain/discomfort, followed by anxiety/depression and mobility impairment, while the dimension with the least problems reported was self-care. Therefore, it is necessary to focus on the physical aspects of pain/discomfort and the mental aspects of anxiety/depression when dealing with HRQoL related to drinking in the general population.

### Limitations

As a cross-sectional design study, we could not investigate the longitudinal effects of drinking patterns on the HRQoL. This study also used self-reporting to measure alcohol consumption; thus, some participants may have over- or underestimated their alcohol intake. In these cases, the association between drinking patterns and HRQoL may be erroneously estimated, because the drinking patterns were not accurately classified. In addition, this study could not control for the occasional drinker bias, as occasional drinkers could not be separated.

## 5. Conclusions

Former drinkers have significant differences in HRQoL compared to never drinkers and low-risk drinkers; thus, studies on drinking and health should be designed to minimize the former drinkers’ bias. When former drinkers were separated from never drinkers and low-risk drinkers to control for misclassification bias, there were gender differences in the beneficial association between alcohol use and HRQoL. Although the misclassification error of the estimation of the association was not valid in men, as there was no significant difference in HRQoL between never drinkers and low-risk and hazardous drinkers, the misclassification error of the estimation of the association was valid in women, as the low-risk drinkers showed better HRQoL than those who did not drink. Therefore, it is expected that this study will be used to propose reinforced national alcohol policies that support moderate alcohol use to some extent. Further, it will be necessary to consider gender characteristics when intervening to improve the HRQoL related to drinking.

## Figures and Tables

**Figure 1 ijerph-17-07758-f001:**
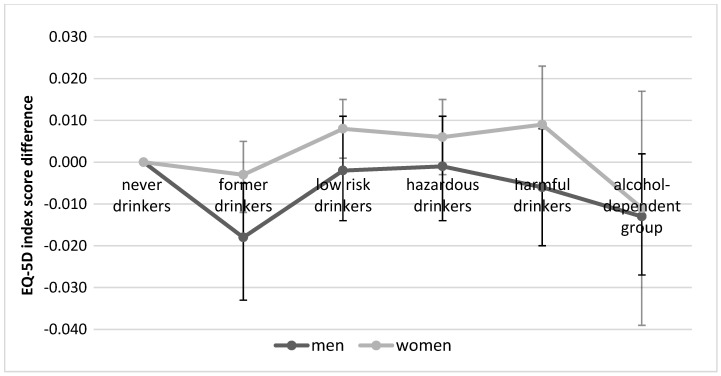
Adjusted EuroQol 5-Dimensions (EQ-5D) index score differences associated with alcohol drinking patterns by gender.

**Table 1 ijerph-17-07758-t001:** Sociodemographic and health-related characteristics of subjects by alcohol drinking patterns.

Characteristics	Categories	N (% *)	Never Drinkers (*n* = 3210)	Former Drinkers (*n* = 3100)	Low Risk Drinkers (*n* = 10,506)	Hazardous Drinkers (*n* = 4018)	Harmful Drinker (*n* = 1141)	Alcohol Dependent Group (*n* = 1080)	*p*
				% * (SE)					
Total		23,055 (100.0)	10.6 (0.3)	11.7 (0.3)	44.8 (0.4)	21.0 (0.4)	5.9 (0.2)	6.0 (0.2)	<0.001
Age (years), Mean (SE)		45.4 (0.3)	58.2 (0.5)	50.5 (0.4)	44.0 (0.2)	40.3 (0.3)	42.8 (0.5)	43.2 (0.5)	<0.001
Gender	Men	9756 (49.2)	19.4 (1.0)	30.6 (1.0)	39.6 (0.6)	74.4 (0.8)	85.9 (1.3)	85.3 (1.3)	<0.001
	Women	13,299 (50.8)	80.6 (1.0)	69.4 (1.0)	60.4 (0.6)	25.6 (0.8)	14.1 (1.3)	14.7 (1.3)	
Economic status	1Q (lowest)	5557 (26.3)	28.0 (1.1)	32.2 (1.1)	24.4 (0.7)	25.5 (1.0)	25.8 (1.8)	28.6 (1.7)	<0.001
	2Q	5771 (25.6)	26.6 (1.0)	26.0 (1.1)	25.7 (0.7)	24.9 (0.9)	23.2 (1.6)	27.3 (1.6)	
	3Q	5740 (24.6)	23.5 (0.9)	22.0 (0.9)	25.2 (0.6)	25.7 (0.9)	26.0 (1.6)	22.1 (1.6)	
	4Q (highest)	5771 (23.5)	21.8 (1.1)	19.8 (0.9)	24.6 (0.7)	23.9 (1.0)	24.9 (1.6)	22.1 (1.5)	
Education level	Elementary school	5821 (18.5)	46.2 (1.3)	26.8 (0.9)	16.1 (0.6)	9.0 (0.5)	9.7 (1.0)	13.3 (1.3)	<0.001
	Middle school	2476 (9.9)	12.7 (0.7)	10.3 (0.7)	9.3 (0.3)	9.0 (0.5)	9.4 (1.0)	11.7 (1.1)	
	High school	7766 (38.9)	23.9 (1.1)	32.4 (1.2)	39.3 (0.7)	45.3 (1.1)	47.0 (1.7)	44.9 (1.8)	
	Higher education	6984 (32.7)	17.2 (1.1)	30.5 (1.1)	35.3 (0.7)	36.7 (1.0)	33.9 (1.6)	30.1 (1.7)	
Marital status	Married	16,806 (67.5)	66.8 (1.2)	72.6 (1.0)	67.7 (0.7)	62.7 (1.1)	70.7 (1.9)	71.3 (1.8)	<0.001
	Never married	3279 (22.3)	7.9 (0.9)	11.3 (0.8)	23.8 (0.7)	31.9 (1.1)	23.6 (1.9)	22.6 (1.7)	
	Bereaved, Divorced, Separated	2942 (10.2)	25.3 (1.0)	16.2 (0.8)	8.5 (0.3)	5.4 (0.4)	5.8 (0.8)	6.0 (0.9)	
Employment	Unemployed	9527 (36.3)	54.2 (1.2)	52.0 (1.1)	38.4 (0.6)	24.8 (0.9)	18.5 (1.5)	17.0 (1.3)	<0.001
	Employed	13,476 (63.7)	45.8 (1.2)	48.0 (1.1)	61.6 (0.6)	75.2 (0.9)	81.5 (1.5)	83.0 (1.3)	
Smoking status	Never smoked	13,725 (54.5)	87.1 (0.8)	70.7 (1.0)	64.8 (0.6)	29.5 (0.9)	15.2 (1.2)	13.9 (1.3)	<0.001
	Ex-smoker	4614 (19.6)	6.4 (0.5)	18.6 (0.8)	17.6 (0.4)	25.8 (0.8)	29.6 (1.6)	28.1 (1.7)	
	Current smoker	4711 (25.9)	6.5 (0.6)	10.7 (0.7)	17.6 (0.5)	44.7 (1.1)	55.2 (1.7)	58.0 (1.9)	
Physical activity	Active	12,405 (52.6)	58.2 (1.2)	57.9 (1.1)	52.2 (0.6)	48.7 (1.0)	49.1 (1.7)	51.9 (1.7)	<0.001
	Inactive	10,596 (47.4)	41.8 (1.2)	42.1 (1.1)	47.8 (0.6)	51.3 (1.0)	50.9 (1.7)	48.1 (1.7)	
Obesity	No	15,575 (67.9)	64.8 (1.1)	68.9 (1.1)	71.6 (0.6)	65.1 (0.9)	60.1 (1.8)	60.9 (1.7)	<0.001
	Yes	7295 (32.1)	35.2 (1.1)	31.1 (1.1)	28.4 (0.6)	34.9 (0.9)	39.9 (1.8)	39.1 (1.7)	
Chronic disease	No	19,976 (90.3)	76.2 (1.0)	84.5 (0.7)	91.2 (0.3)	95.8 (0.4)	96.6 (0.5)	95.0 (0.8)	<0.001
	Yes	3079 (9.7)	23.8 (1.0)	15.5 (0.7)	8.8 (0.3)	4.2 (0.4)	3.4 (0.5)	5.0 (0.8)	

* weighted %, SE: standard error.

**Table 2 ijerph-17-07758-t002:** Adjusted odds for impaired health-related quality of life associated with alcohol drinking patterns.

Health-Related Quality of Life		Never Drinkers	Former Drinkers	Low Risk Drinkers	Hazardous Drinkers	Harmful Drinkers	Alcohol Dependent Group	*p* *
		Adjusted OR (95% CI)						
EQ-5D Index (L2 and L3)								
Mobility impairment	Model 1	1	1.14 (0.96–1.35)	0.92 (0.79–1.07)	0.82 (0.66–1.02)	0.96 (0.70–1.31)	1.36 (0.99–1.87)	0.001
	Model 2	1	1.12 (0.93–1.35)	0.95 (0.81–1.12)	0.81 (0.64–1.03)	0.95 (0.69–1.31)	1.28 (0.92–1.76)	0.007
Self-care impairment	Model 1	1	1.06 (0.85–1.31)	0.78 (0.64–0.95) *	0.69 (0.48–0.97) *	1.04 (0.60–1.79)	1.14 (0.70–1.87)	0.010
	Model 2	1	1.04 (0.83–1.31)	0.79 (0.64–0.98) *	0.67 (0.47–0.96) *	1.03 (0.60–1.77)	1.07 (0.65–1.77)	0.027
Usual activities impairment	Model 1	1	1.25 (1.04–1.50) *	0.87 (0.75–1.01)	0.82 (0.64–1.05)	1.28 (0.88–1.88)	1.29 (0.91–1.83)	0.000
	Model 2	1	1.23 (1.02–1.49) *	0.88 (0.75–1.03)	0.81 (0.63–1.04)	1.28 (0.87–1.87)	1.19 (0.83–1.70)	0.000
Pain/discomfort	Model 1	1	1.24 (1.08–1.43) *	0.99 (0.89–1.11)	1.03 (0.88–1.20)	1.04 (0.82–1.31)	1.37 (1.10–1.70) *	0.001
	Model 2	1	1.24 (1.07–1.44) *	1.01 (0.89–1.14)	1.01 (0.86–1.19)	1.01 (0.80–1.28)	1.31 (1.05–1.65) *	0.004
Anxiety/ depression	Model 1	1	1.32 (1.11–1.58) *	1.16 (0.99–1.36)	1.39 (1.12–1.73) *	1.49 (1.09–2.06) *	2.63 (2.03–3.42) *	0.000
	Model 2	1	1.30 (1.09–1.56) *	1.15 (0.98–1.35)	1.29 (1.04–1.61) *	1.42 (1.03–1.94) *	2.42 (1.86–3.14) *	0.000

L2: some or moderate problem and L3: extreme problems. OR: odds ratio and CI: confidence interval. Odds ratio of experiencing “some-to-extreme” problems across the EuroQol 5-Dimensions (EQ-5D) associated with drinking patterns. Model 1: Adjusted for age (y), gender, economic status, education level, marital status, and employment status. Model 2: Adjusted for the factors listed in Model 1, in addition to smoking status, physical activity, obesity, and chronic disease. * *p*-values were corrected by Bonferroni’s method due to multiple testing.

**Table 3 ijerph-17-07758-t003:** Means of the EQ-5D index score according to alcohol drinking patterns by gender.

Drinking Patterns	Male (*n* = 9759)					Female (*n* = 13,299)				
	% * (SE)	Unadjusted EQ-5D Index		Adjusted EQ-5D Index		% (SE)	Unadjusted EQ-5D Index		Adjusted EQ-5D Index	
		Mean (SE)	*p* *	Mean (SE)	*p* *		Mean (SE)	*p* *	Mean (SE)	*p* *
Never drinkers	4.2 (0.3)	0.953 (0.006)	ref	0.913 (0.007)	ref	16.8 (0.4)	0.882 (0.004)	ref	0.880 (0.005)	ref
Former drinkers	7.3 (0.3)	0.931 (0.006)	0.034	0.895 (0.006)	0.065	15.7 (0.4)	0.912 (0.003)	<0.001	0.877 (0.004)	1.000
Low risk drinkers	36.2 (0.6)	0.965 (0.001)	0.275	0.912 (0.005)	1.000	53.4 (0.6)	0.948 (0.001)	<0.001	0.888 (0.003)	0.119
Hazardous drinkers	31.6 (0.6)	0.972 (0.001)	0.016	0.913 (0.005)	1.000	10.6 (0.4)	0.955 (0.003)	<0.001	0.886 (0.004)	1.000
Harmful drinkers	10.4 (0.5)	0.967 (0.003)	0.268	0.908 (0.005)	1.000	1.6 (0.2)	0.959 (0.007)	<0.001	0.889 (0.007)	1.000
Alcohol dependent group	10.3 (0.5)	0.955 (0.004)	1.000	0.901 (0.006)	0.434	1.8 (0.2)	0.937 (0.014)	0.001	0.870 (0.013)	1.000

Adjusted for age (y), household level, education level, marital status, employed status, smoking status, physical activity, obesity, and chronic disease. CI: confidence interval. * *p*-values were corrected by Bonferroni’s method due to multiple testing.

**Table 4 ijerph-17-07758-t004:** Adjusted EQ-5D index score differences associated with alcohol consumption patterns by gender.

	Male (*n* = 9759)		Female (*n* = 13,299)	
	Estimate (95% CI)	*p* *	Estimate (95% CI)	*p* *
Never drinkers	ref		ref	
Former drinkers	−0.018 (−0.033 to −0.004)	0.013	−0.003 (−0.012 to 0.005)	0.453
Low risk drinkers	−0.002 (−0.014 to 0.011)	0.798	0.008 (0.001 to 0.015)	0.024
Hazardous drinkers	−0.001 (−0.014 to 0.011)	0.854	0.004 (−0.005 to 0.013)	0.422
Harmful drinkers	−0.006 (−0.020 to 0.008)	0.415	0.009 (−0.006 to 0.023)	0.262
Alcohol dependent group	−0.013 (−0.027 to 0.002)	0.087	−0.011 (−0.039 to 0.017)	0.446

Adjusted for age (y), household level, education level, marital status, employed status, smoking status, physical activity, obesity, and chronic disease. CI: confidence interval. * *p*-values were corrected by Bonferroni’s method due to multiple testing.

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
