# Peer review of "The Association between Alcohol Drinking Patterns and Health-Related Quality of Life in the Korean Adult Population: Effects of Misclassification Error on Estimation of Association"

_ijerph, 2020, doi:10.3390/ijerph17217758_

Round 1

Reviewer 1 Report

This manuscript is based on useful data and has an interesting perspective. However, much can be done to narrow down the focus. I will detail how I think this can be done, as well as offering some minor comments.

The abstract states the purpose as "...to validate the protective effect of alcohol use on HRQoL..." However, QoL is not a risk, nor is its absence. I would restate this as "testing whether the J-shaped association between alcohol use and HRQoL holds when separating former drinkers from never drinkers" or something along those lines.

Similarly, a positive correlation between a health factor and a QoL score does not indicate causality (any more than a negative would). Therefore, the term "protective" is mis-leading and should be avoided.

The abbreviation GBD should be written out the first time it occurs, as well as in the references list.

This statement does not quite follow: "Although men experience more deaths
88 and disabilities due to consuming more alcohol than women [2], they have a higher level of HRQoL than women [9,10,14,24], so there may be gender differences in the association between drinking and HRQoL." As the second part of the statement does not say anything about alcohol, the two statements are not logically connected. While the authors can make the connection, they need to rephrase it, and make sure that it does not over-state the strength of the statement. 

I was a bit puzzled that the purpose statement of the study did not mention the J- or U-shaped curves mentioned in the introduction. If they are not relevant for the purpose of the study, they should be excluded (and dropped from the analyses), if they are important they should be part of the purpose.

"Patients who were diagnosed with one or more of the following diseases were classified as being in the chronic disease morbidity
group.
" (p. 4) - this is not followed by diagnoses, the next words are the heading "Data analysis".

The authors state that they collapsed the two values for each EQ-5D item and conducted a logistic regression, but did not state why they did this rather than, say, use ordinal logistic regression. I am not saying that what they did is wrong, but only that there must be some reason - even if that is simply that this is how other studies have done.

The manuscript contains a very large number of signficance tests. The data analysis plan does not mention anything about adjusting for this fact, but given that there are a large number of drinking pattern being compared in regards to a large number of QOL variables, and even twice that number because the analyses are stratified by gender, some adjustment should be made. 

In the results section the authors refer to people as "alcohol dependence" (e.g., "alcohol dependence (OR, 2.42; 95%CI, 1.86. 3.14) showed significantly higher ORs for anxiety/depression compared to never drinkers"). Please change into "people with alcohol dependence". Also consider person-first language throughout (e.g., "respondents who were former drinkers...").

Table 2 is a really pain to read, because of the formatting of the columns. 

Table 2 and 3 test essentially the same relationships. I suggest moving one of them to an appendix.

What does boldface mean in Table 4?

The discussion should start with evaluating the findings in light of the hypotheses, rather than comparing a figure that is not central to the study's hypotheses with figures from another survey. 

The authors put a long paragraph into the discussion commenting on the frequency of various types of impairment that do not differ between drinking groups. I do not see the usefulness of that. More people report pain than problems moving about. Yes, true indeed, but more respondents classified as never-drinkers also reported pain than any other type of impairment. Well, that says nothing about the topic of the paper, and may eventually say more about the wording of the EQ-5D than about the types of impairment - there is simply no way of knowing. 

Author Response

Thank you very much for reviewing this manuscript. I have attempted to address the reviewers’ comments and improve the manuscript. I have included point-by-point responses to each comment in this re-submission. 

Reviewer 2 Report

  • Summary

    • The authors are providing insight into the relationship between alcohol abuse and health-related Quality of Life (hrQoL) as a meta-analysis of results from the Korean population. They describe the necessity to correct for confounding life parameters such as alcohol abuse in a period of life prior to the investigation and sex-related biases. The arguments are out in perspective very well, while the corrected analysis is of very high statistical quality, giving better insight into the cohorts.
    • The cut-off selections for cohort definition using the AUDIT are in agreement with general literature, while hrQoL assessment using the EQ-5D also was on point. All resulting population statistics are scaling as expected and are not necessarily limited to the Korean population, giving the paper an overarching confirmatory scope.
    • The paper confirms interesting differences between drinking patterns and lifetime statistics, thereby offering perspectives for future assessments of associated factors, such as age and drinking status. Most importantly, the importance of discriminating never-drinkers from former drinkers and low-risk drinkers is highlighted by this analysis, pointing to a major issue with previous studies.
  • Major remarks

    • Please provide the original output of the regression models in the supplements. Without this information, it is hard to read the relationships and setups from the statistics section. Therefore, it is vital to control the validity of the analysis. All the chosen methods appear valid, but the setup is too complex to be explained in the narrated form.
  • Minor remarks

    • line 159: linear regression
    • line 329: Health

Author Response

(The authors gave the same response as above.)

Round 2

Reviewer 1 Report

The authors have addressed many of my comments, but a new terminology has emerged, "the estimation of association." I can only guess what the authors mean by that term, and what they mean be the estimation being valid in women but not in men. But if I try to read between the lines, it seems that what the authors mean is that the (male) former drinkers influence the results, because their QOL is worse than that of all men. Thus, my initial understanding that this was linked with the J-shaped curve in a major way seems to be wrong, and I would recommend that the authors remove references to that, now that I know what this is actually about.

Author Response

Dear reviewer

Thank you for the opportunity to revise our manuscript. We appreciate the careful review and constructive suggestions. It is our belief that the manuscript is substantially improved after making the suggested edits.  
